# Clinical Aspects of Janus Kinase (JAK) Inhibitors in the Cardiovascular System in Patients with Rheumatoid Arthritis

**DOI:** 10.3390/ijms21197390

**Published:** 2020-10-07

**Authors:** Przemysław J. Kotyla, Md Asiful Islam, Małgorzata Engelmann

**Affiliations:** 1Department of Internal Medicine, Rheumatology and Clinical Immunology, Faculty in Katowice, Medical University of Silesia, 40-635 Katowice, Poland; 2Department of Haematology, School of Medical Sciences, Universiti Sains Malaysia, Kubang Kerian 16150, Kelantan, Malaysia; 3Department of Physiotherapy in Internal Medicine, Academy of Physical Education in Katowice, 40-065 Katowice, Poland; m.engelmann@awf.katowice.pl

**Keywords:** rheumatoid arthritis, JAK/STAT, Janus kinase inhibitors, cardiovascular system, heart failure, thromboembolic, lipid profile disturbances, cytokines

## Abstract

Janus kinase (JAK) inhibitors, a novel class of targeted synthetic disease-modifying antirheumatic drugs (DMARDs), have shown their safety and efficacy in rheumatoid arthritis (RA) and are being intensively tested in other autoimmune and inflammatory disorders. Targeting several cytokines with a single small compound leads to blocking the physiological response of hundreds of genes, thereby providing the background to stabilize the immune response. Unfortunately, blocking many cytokines with a single drug may also bring some negative consequences. In this review, we focused on the activity of JAK inhibitors in the cardiovascular system of patients with RA. Special emphasis was put on the modification of heart performance, progression of atherosclerosis, lipid profile disturbance, and risk of thromboembolic complications. We also discussed potential pathophysiological mechanisms that may be responsible for such JAK inhibitor-associated side effects.

## 1. Introduction

Rheumatoid arthritis (RA) is the most common form of inflammatory polyarthropathies, affecting approximately 1% of the population worldwide. When the disease is not treated or treated insufficiently, it ultimately leads to permanent joint destruction, and subsequent disability [1]. In addition, as a member of systemic connective tissue diseases, RA is linked to an increased risk of internal organ involvements and systemic complications with premature atherosclerosis being the most important one. In the last two decades, significant therapeutic progress has been made and new therapeutic strategies implemented, resulting in better disease control and leading to a sustained remission or at least low disease activity. The new approach to RA treatment is based on the early introduction of synthetic disease-modifying antirheumatic drugs (DMARDs), mainly methotrexate (MTX), and is aimed to achieve remission or low disease activity. Unfortunately, only part of the patients responds well to such a treatment.

At the end of the last century, the therapeutic regimens for treating RA widened and new therapeutic strategies were implemented to the treatment repertoire. The new group of therapeutic molecules called biological DMARDs (bDMARDs) or simply biologics were introduced to the common clinical practice. The mode of action of biologics is based on blocking the inflammatory cytokines, depleting the population of antibody producing B-cells, and interfering in the co-stimulation of immunocompetent cells. This new approach revolutionized the treatment of RA. However, a substantial portion of patients still do not respond to such treatment. The limitation of these therapeutic strategies is the fact that biologics, which are high molecular weight compounds, are highly immunogenic and often produce adverse drug reactions such as tuberculosis, heart failure, neuropathies, and others [2,3,4]. Also of note is the fact that they are given parenterally, are expensive in manufacturing, and difficult to handle. 

The progress made in immunology over the last 20 years contributed to the better understanding of the mechanisms of autoimmune diseases that translated directly to the development of new therapeutic approaches.

Among them, the Janus kinases/signal transducers and activators of transcription (JAK/STAT) pathway attracts high interest, and offers blockade of several cytokines with one small synthetic compound [5,6,7,8]. In line with the discovery of the JAK/STAT pathway, several synthetic compounds which are able to block this pathway have been developed [9]. This group of new synthetic DMARDs is called JAK inhibitors and indeed offer the blockade of many cytokines with one small compound (Table 1). Treatment with JAK inhibitors proven to be efficacious and relatively safe, and is recognized as equal and even superior to the conventional biologics [10]. This approach however may not be entirely free of risk of developing side effects. Blockade of several cytokines and interferons (IFNs) may bring many pathophysiological consequences as JAK/STAT pathway is deeply involved in several largely unknown regulatory networks and reduction of cytokine synthesis may contribute to the reduction of inflammatory process on one side in addition of dysregulating immune, cardiovascular, and nervous systems. As the example may serve tumor necrosis factor (TNF)-α inhibition that is responsible for worsening of the heart function in patients with congestive heart failure, lipid profile disturbances as the result of interleukin (IL)-6 inhibition, or increased risk of thromboembolism in the course of JAK inhibitor administration.

In this review, we tried to discuss the pathophysiological mechanism that may explain adverse drug reactions in the cardiovascular system during treatment with JAK inhibitors. Special emphasis was put on the role of cytokines blocked during the treatment with JAK inhibitors and their pathophysiological impacts on the functioning of the cardiovascular system.

## 2. JAK-STAT Pathway

JAKs are enzymes belonging to the tyrosine kinase family enzymes and their main function is to phosphorylate tyrosine residues to activate downstream signaling proteins and evoke physiological functions. When activated, they transfer extracellular signals provided by growth factors, cytokines, and chemokines that translate directly to the change of DNA transcription with the subsequent translation of several proteins. At the moment, four JAKs have been identified in mammalians (JAK1, JAK2, JAK3, and TYK2) which are specifically attached to receptors [11]. Activation of one specific JAK by the ligand-receptor can be recognized by the various receptor-ligand complexes as the one specific JAK and could be activated by several cytokines. *JAK1*, *JAK2*, and *TYK2* are expressed by many cells, contrary to this; hematopoietic, myeloid, and lymphoid cells express JAK3 [12]. The activation of JAK is a multi-step process. After a ligand is ligated to the receptor, the receptor’s subunits dimerize and form an active receptor which is able to activate receptor-associated JAK [13]. Active phosphorylated JAK then phosphorylates tyrosine residues in the cytoplasmic part of the receptor enabling creation of docking sites for STAT. STATs are DNA-binding proteins which, when phosphorylated (activated), dimerize and translocate to the nucleus, followed by regulation of gene expression. It makes STATs the second key player in the transmission of signal. Currently, seven STAT proteins have been identified. The JAK/STAT system is responsible to transmit signals of more than 50 ligands and is recognized as one of the central communication systems of the immune response [14]. Active STATs then translocate to the nucleus where they interact with DNA regulatory elements, changing the expression of related genes [15,16]. 

JAK and more precisely JAK/STAT system is responsible for transmitting the signals provided by the wide spectrum of cytokines, which are ligands for class I and class II receptors. These receptors are protein complexes expressed on the surface of cells. They are built as one to four receptor chains. The typical structure of the receptor is formed from extracellular cytokine R homology domain (CHD) and a sequence acting as cytokine binding site. Slight structural differences in the CHD cytokine receptors enable to distinguish class I or class II family receptors [17]. Class I receptor family interacts with four cytokine families - gamma chains (γc), beta chains (βc), cytokines that utilize gp130 protein, and ILs that interact with a receptor’s common subunit p40. Presence of γc in the receptor enables to interact with IL-2, IL-4, IL-7, IL-9, IL-15, and IL-21 [18], since βc is responsible for transducing the signals provided by granulocyte-macrophage colony-stimulating factor (GM-CSF), IL-3, and IL-5. Several cytokines utilize gp130 protein as a component of their receptor. This cytokine subfamily consists of IL-6, IL-11, IL-27, and IL-31. This receptor complex is also used by ciliary neurotrophic growth factor, oncostatin M, cardiotrophin1 and cardiotrophin-like cytokine factor 1 [19,20,21]. Recently, two other cytokines namely IL-35 and IL-39 have been added to gp130 family due to the fact that they use gp130 as a signal transmitting unit in the receptor complex [22,23]. The last cytokine family that utilize class I receptor consists of IL-12 and IL-23 receptors for heterodimeric cytokines that share the common subunit p40 [24,25]. Several hormone-like cytokines as growth hormone, leptin, erythropoietin, and thrombopoietin also transmit signals via the class I receptor. Plethora of cytokines, chemokines, and growth factors makes class I receptors the real crossroad of immune response, metabolism growth, tissue development, and indicates how important modulation of this pathway is. βc-the family of class II cytokines comprise a large group of signaling molecules. The most important members of this class are type I, II, and III IFNs (IFN-α, IFN-β, IFN- γ, IL-28, and IL-29) and cytokines belonging to IL-10-related family (IL-10, IL-19, IL-20, IL-22, IL-24, and IL-26) [17].

The transmission of signals from receptors requires two molecules of JAKs. JAK1 transmits signals provided by IL-6, IL-10, IL-11, IL-19, IL-20, and IL-22, and IFN-α, IFN-β, and IFN-γ, since JAK2 activation is responsible for the signaling of hormone-like cytokines-erythropoietin, thrombopoietin, growth hormone, GM-CSF, IL-3, and IL-5. [26]. JAK3 is exceptional member of JAK family which transmits signals as heterodimer of JAK1 and JAK3 molecules and is primarily expressed on hematopoietic cells. Attached to γ-chain transmits signals from IL-2, IL-4, IL-7, IL-9, IL-15, and IL-21 [26]. The last member of JAK family TYK2 facilitates signaling for IL-12, IL-23, and type I IFNs [27]. TYK2 creates heterodimers with either JAK1 or JAK2 [27]. Signaling the various cytokines by specific pairs of JAKs at least potentially creates a chance to target (inhibit) narrow branch of cytokines. However, one should remember that, contrary to biologic-targeted therapy, when one drug blocks only one cytokine, JAK inhibitor blocks many cytokines which utilize the same type of receptor.

## 3. The Role of JAK/STAT Pathway in Immunity and Autoimmunity

The results from many studies performed in the last two decades confirmed the involvement of the JAK/STAT pathway in several diseases associated with inflammation, cancer, immunity, and immune deficiency. This is not surprising as JAK mutations have been identified in immune deficiency syndromes including severe combined immune deficiency (SCID), in hematologic malignancies (leukemias and lymphomas) [28], and also in autoimmunity (hyper IgE-Job’s syndrome) [29]. Some of the mutations in the JAK/STAT pathway directly increase the risk of developing well characterized autoimmune disorders like inflammatory bowel disease, psoriasis, ankylosing spondylitis, Behçet’s disease [30,31,32], RA, Sjögren syndrome, or systemic lupus erythematosus (SLE) [33,34]. These findings underline the role of cytokine mediated regulation that affects such important pathophysiological processes as IFN-mediated immunity [35,36], T- and NK-cell-based immune response, regulation of function of lymphocytes, hematopoiesis, and nerve development. Recently, the role of IFNs in the development of several autoimmune disorders has been confirmed. In line with it, the term IFN signature has been coined indicating the special role of IFNs in the devolved of autoimmunity, specifically a prominent increase in the expression of type I IFN-regulated genes. This is especially a fact as far as SLE, inflammatory myopathies, and systemic sclerosis are concerned [37,38,39,40,41].

SLE is an autoimmune disease with a complex immunopathogenesis where B-cells have been implicated in humoral abnormalities and a prominent type I IFN signature is found in blood of majority of the SLE patients [42]. As the JAK/STAT cascade was identified to be responsible for the signal transduction from the activated IFN receptor to the nucleus, any disturbance in activity of this pathway may lead to the disease development. Indeed, the study on human lupus nephritis (LN) by Arakawa et al. [43] observed increased glomerular staining of STAT3 in renal biopsies of LN patients. Moreover, in patients with different types of glomerulonephritides, STAT3 activation highly correlated with glomerular and tubulointerstitial cell proliferation, interstitial fibrosis, and the level of renal injury. Obviously, the role of cytokines in the development of autoimmune diseases is not limited to IFNs. All known and perhaps not already known cytokines and chemokines create unique networks of self-interactions, activation, and regulatory loops. Any disturbance in this precise universe results directly to the development of autoimmunity, malignancy, allergy, or immunodeficiency. In addition, the second main player in autoimmunity, namely B-cells, are at least partially dependent upon cytokine stimulation utilizing the JAK/STAT system for signal transmission [44].

In the last four decades, the importance of cytokines in autoimmune diseases, as an executive arm of autoimmunity has been established. As the JAK/STAT pathway is one of the three most important signaling pathways in the cell, targeting of JAKs appears to be a rational strategy to stop the development of the diseases at a very early stage. In line with it, a great number of JAK inhibitors in various stages of preclinical development are being tested in clinical trials, and some of them have already been approved for the treatment. 

## 4. JAK Inhibitors and Rheumatoid Arthritis

RA is a chronic inflammatory polyarthropathy characterized by the symmetrical involvement of peripheral joints, internal organ involvements, and systemic symptoms [1]. The etiology of the disease is not precisely understood but at the current level of knowledge and our understanding based on the disease mechanism, the pathogenesis is believed to be the mosaics of environmental, genetic, and lifestyle-related factors. All these factors when working together contribute to the aberrant immunological response and create the autoimmune reactions. Targeting many pro-inflammatory cytokines with a single small molecule created a unique opportunity to block several signaling pathways involved in the development of autoimmune diseases including RA. The obvious background to target JAK was to reduce the level of IL-6, one of the pivotal cytokines in RA. The other cytokines, although not directly involved into the pathogenesis of RA, create permissive background for inflammatory response contributing to the development of cellular response [45], including the Th1, Th2, and Th17 cells, which are directly involved in the development of autoimmune and inflammatory disorders [46]. Understanding the role of JAK in the pathogenesis of various autoimmune disorders led to the synthesis of several JAK inhibitors. Followed the approval of the first JAK inhibitor, tofacitinib, in the treatment of RA in 2012 and in 2017 in the USA and Europe, respectively, baricitinib and upadacitinib were subsequently approved for RA in DMARD failure patients.

Based on the selectivity of a given JAK molecule, JAK inhibitors are commonly subcategorized as first and next generation. First generation of JAK inhibitors (i.e., tofacitinib, baricitinib, and peficitinib, all approved for RA in Japan) block two or more JAK molecules resulting in inhibition of several cytokines. Consequently, inhibiting a JAK molecule may block more than one pathway, which may in part explain both the drug efficacy and some of the adverse effects observed with JAK inhibitor treatment [47]. In contrast, next generation of JAK inhibitors (i.e., upadacitinib and filgotinib, which are not approved for RA yet) is characterized by high selectivity, and thus administration of inhibitors blocks only one specific JAK molecule and selectively inhibits signal from one or limited number of cytokines. This provide the precise mechanism enabling to target one cytokine with one drug.

The current strategy for the treatment of RA is based on early starting classic synthetic DMARDs (csDMARDs), mainly methotrexate (which is still recognized as an anchor drug in the treatment regimen), administered alone or in combination with glucocorticosteroids. In case of MTX failure or intolerance, other csDMARDs (i.e., sulfasalazine or leflunomide) are available. The other group of therapeutic agents including bDMARDs or targeted synthetic DMARDs (tsDMARDs) with JAK inhibitors being the most important representative.

Blocking several cytokines with a single small molecule was a successful approach. In many clinical trials, JAK inhibitors were proven to be equal to bDMARDs [10,48,49]. Two studies designed as non-inferiority trials have shown statistical superiority of baricitinib or upadacitinib compared with adalimumab (all in combination with MTX) [50,51]. However, a third study using tofacitinib+MTX did not show such efficacy [52]. Therefore, the European League Against Rheumatism (EULAR) task force decided that clinical significance is too low to prefer tsDMARDs over bDMARDs. In line with this conclusion, current EULAR recommendation indicates adding bDMARD or a tsDMARD to the treatment regimen when the treatment target is not achieved with the first csDMARD strategy and poor prognostic factors are observed. Moreover, EULAR task force revised the preference of bDMARDs over tsDMARDs (proposed in earlier recommendation) because of new evidence regarding the successful long-term efficacy and safety of JAK inhibitors [53,54,55]. Currently, four JAK inhibitors are approved for the treatment of RA, namely tofacitinib, baricitinib, upadacitinib, and peficitinib (approved for RA in Japan), but many other compounds are currently tested in RA and other autoimmune diseases [56].

## 5. Safety Issues of JAK Inhibitors

Blocking several cytokines with one small molecule may potentially bring many pathophysiological consequences. It is especially true when we consider how blocking the different pathways translates to change the activity of various, sometime critical body systems. Firstly, targeting JAK3 attached to γ-cytokines leads to impairment of signaling via IL-2, IL-4, IL-7, IL-9, IL-15, and IL-21 (Figure 1). This is of the special importance as those cytokines are responsible for proper T-cell development and immunoglobulin synthesis. JAK3 blockade resembles severe immunodeficiency syndrome, with a switch off mutation in γ-chain resulting in X-linked SCID [57]. Therefore, infection, especially viral infections attracted the main attention of researches. Targeting JAK and blocking transmission of several cytokines may also lead to some safety issues in the cardiovascular system.

### 5.1. Heart Failure 

JAK/STAT pathway transmits not only inflammatory signals, but also is deeply involved in the proper functioning of many systems of the body including the cardiovascular system. In neonatal rat, cardiocytes angiotensin II induces JAK2 phosphorylation and this process is critically depended on reactive oxygen species generation via membrane-bound NADP-oxidase. This may offer an important link between high glucose levels and glucose-dependent angiotensin II-mediated phosphorylation of JAK2. Of note is the fact that cardiac hypertrophy in non-failing heart is dependent on JAK2 phosphorylation [58]. This may potentially offer therapeutic approach in patients with RA, where insulin resistance and latent diabetes contribute to cardiac hypertrophy and heart failure. On the other hand, it was shown in a mice model that gp130 receptor and gp130 cytokines may offer survival pathway in transition to the heart failure (Figure 1) [59]. Thus, switching off this pathway via JAK inhibition ameliorates compensatory effect upon heart at risk for even failure. The pivotal role of cytokine that utilized gp130 receptor, namely IL-6 is still the matter of controversy. Contrary to TNF, which is known to contribute to heart insufficiency, the role of IL-6 is widely unknown. Quite recently, Hengdong et al. in their meta-analysis showed increased level of IL-6 was independently associated with higher risk of major adverse cardiovascular events (MACE), cardiovascular and all-causes of mortality in patients with acute coronary syndromes [60]. Although at this moment, it is not clear if IL-6 is the only valuable biomarker of heart failure or it is only pathophysiologically involved in heart failure. More information was obtained from the study of Liangjie et al. who assayed IL-6 and IL-17 in patients who underwent cardiac catherization. In this study, IL-6 and IL-17 levels were correlated with the levels of fibrotic parameters indicating the role of both cytokines in the development of heart insufficiency [61]. Recently, some indirect data suggested that targeting IL-1 with subsequent reduction of IL-6 exerted a significant effect on the primary cardiovascular end point [62]. IL-6 is the central inflammatory cytokine, and together with its downstream inflammatory biomarker CRP is linked to high cardiovascular risk [63]. 

Despite numerous experimental and clinical studies, the role of IL-6 on the development of heart failure has not yet been fully elucidated. It is believed that high concentration of IL-6 can lead at least partially to the development of heart failure [64] and may serve as an indicator of worse prognosis in patients with cardiovascular diseases [65]. Targeting JAK with subsequent blockade of gp130 mediated pathway may potentially facilitate to stabilize the heart function. At this moment, it is unclear whether this improvement is due to the direct reduction of IL-6 or this process is mediated via limitation of an inflammatory state. Blocking the JAK/STAT pathway may also bring many negative consequences. It is well known that activation of the JAK2/STAT3 pathway protects the myocardium against ischemia/reperfusion injury and inhibits apoptosis of the coronary artery endothelial cells [66], thus provide mechanism that are far beyond only cytokine signaling [67]. Contrary to this, in another study, inhibition of JAK2/STAT partially attenuated the pro-apoptotic effect of IL-23 (a member of IL-12 family) upon cardiomyocytes [68]. Therefore, it is suggested that, in these circumstances, IL-23 promotes the activation of JAK2/STAT pathway and enhances the expression of IL-17, the cytokine deeply involved in myocardial ischemia/reperfusion injury (Figure 1) [69]. It is an identical fact when the other member from this cytokine family, IL-12, is concerned. 

Previous studies reported that plasma IL-12 concentrations were significantly increased in many types of atherosclerosis and atherosclerotic cardiovascular disease. At this moment, it is however unclear whether inhibition of signaling pathways via JAK/STAT system may attenuate the harmful effect of IL-12 upon the heart, which is not a surprising finding. The inhibition of JAK results in the reduced expression of many cytokines belonging to the various families. Moreover, even in the same cytokine family, some of them exert pro-inflammatory response since the other cytokines are recognized as anti-inflammatory ones. IL12 and IL-23 demonstrate strong pro-inflammatory properties. However, the other member of this family, IL-35, exerts anti-inflammatory potentials. Therefore, the direct effect of the inhibition of the JAK/STAT pathway is dependent on what cytokines are predominantly blocked when inhibition of the JAK/STAT pathway occurs (Figure 2).

More data were provided from clinical studies where cardiovascular risk was assessed. Those studies demonstrated a low-incidence rate of MACE in RA patients, suggesting a good cardiovascular profile of JAK inhibitors [70]. Recently, these findings were substantiated by the first meta-analysis exploring the relationship between JAK inhibitor treatments and cardiovascular risks. According to the data from this study, short-term treatment with JAK inhibitor does not increase the risk of cardiovascular events when compared to placebo. Furthermore, with the exception of baricitinib, tofacitinib in both 5 and 10 mg doses and upadacitinib (15 mg and 30 mg doses) appeared to be equally safe [71]. Same conclusions were observed from analysis of clinical trials with tofacitinib [72]. The post-hoc analysis comprised in total eight trails - six phase III and long-term extension, respectively. The authors focused on MACE defined as any myocardial infarction, cerebrovascular event (i.e., stroke), or cardiovascular death (defined as death caused by coronary, cerebrovascular, or cardiac events) incidence in a large cohort of 4076 patients representing a total of 12,932 patient-years of tofacitinib exposure. In the study, MACE incidences were linked with older age, longer disease duration, higher mean body mass index, diabetes mellitus, hypertension, and lipid profile changes (higher total low-density lipoprotein (LDL), lower high-density lipoprotein (HDL) cholesterol, triglycerides, and higher total cholesterol to LDL ratio). Contrary to the previous studies linking disease activity with increased risk of poor cardiovascular outcome, in this study, disease activity parameters and inflammatory measured as baseline disease activity and inflammation measures were not significantly associated with MACE [72].

### 5.2. Lipid Profile

RA is linked with increased risk of cardiovascular events. In general population, the role-playing risk factors for atherosclerosis and poor cardiovascular outcome are obesity, sedentary lifestyle, smoking, and lipid profile disturbances. However, a significantly higher risk of developing cardiovascular events in RA patients cannot be explained by the presence of traditional risk factors alone. For many years, RA and other inflammatory conditions are recognized as the independent risk factors. In line with this, hampering the disease activity may add extra beneficial effect on the cardiovascular system.

Patients with RA often show lipid paradox, having lower total cholesterol and its subfractions as compared to unaffected population [73,74]. The most common explanation of this phenomena is suppression of cholesterol synthesis by inflammatory processes. Indeed, there is a strong link between C-reactive protein (a biomarker of inflammation) and circulating lipid levels [74]. Furthermore, treatment of RA may also impact lipid profile and some DMARDs exert the potentials to increase serum LDL and HDL cholesterol levels [75]. These phenomena cannot be explained only by the reduction of inflammation, since impact of various DMARDs on the lipid profile is variable, in spite of similar reductions in disease activity and systemic inflammatory parameters [76,77]. Followed observation on cholesterol increase in tocilizumab-treated RA patients, the interest on impact of various biologic and non-biologic DMARDs increased significantly. tocilizumab, an IL-6 receptor inhibitor, increases circulating LDL levels [77], but no increased risk of major cardiovascular events was observed [78,79]. It is established that tocilizumab reduces the LDL hypercatabolic state and diminishes the expression of LDL receptor on hepatocytes via a proprotein convertase subtilisin/kexin type-9-mediated mechanism [80,81]. Quite recently, Greco et al. showed that treatment with tocilizumab improves the activity of scavenger receptor class B member 1 and ATP binding cassette-G1 (ABCG1) which directly leads to favorable modifications of lipoprotein composition and functions in contributing to the reduced cardiovascular risk in tocilizumab-treated patients [82]. Therefore, contrary to the general population, changes in LDL composition are not translated to increased cardiovascular risk in patients with RA.

#### 5.2.1. JAK Inhibitors and Lipid Profile

In the rabbit model, treatment with tofacitinib decreased systemic and synovial inflammation and increased circulating lipid levels have been observed. In this study, it failed to modify synovial macrophage density, however it reduced the lipid content within synovial macrophages. The study also confirmed the role IFN-γ in formation of foam cells, representing the key element for development of atherosclerosis. Inhibition of JAK/STAT pathway by tofacitinib contributed to lipid release via enhanced expression of cellular liver X receptor α and ATP-binding cassette transporter (ABCA1) synthesis [83]. Similarities in mechanisms of action in those two studies may suggest the role of IL-6 as a mediator of changes in lipid metabolism. The final effect may therefore be achieved by either IL-6 or JAK blockade (transmitting signals form IL-6-dependent receptor). Indeed, in an ex-vivo experiment, it was shown that tofacitinib decreased the expression of the IL-6 gene, having instead a variable effect on that of IL-8, TNF-α, and IL-10 genes [84]. Specifically, with ablation of JAK1 tofacitinib reduced signaling via IFN thus reducing TNF-α synthesis within macrophages. In a study with Chinese RA patients, the serum levels of TNF-α, IL-17, IL-6, and IFN-γ significantly decreased after the treatment with tofacitinib, parallel to an increase of IL-35, with subsequent T-reg lymphocyte response [85].

The role of IFN-γ in the modification of blood lipoproteins and promoting atherosclerosis development is postulated for many years. At the current level of knowledge, IFN signaling via JAK/STAT pathway regulates more than 2300 genes [86]. Signaling via IFN brings many pathophysiological consequences. Being a key element in immune response, IFN is also engaged in lipid metabolism and atherosclerosis development (Figure 3). 

Specifically, it can induce oxidative stress, promote foam cell accumulation, stimulate smooth muscle cell proliferation and migration into the arterial intima, enhance platelet-derived growth factor expression, and destabilize plaque [87]. This makes the IFN and STAT/JAK pathway the potential target to treat atherosclerosis. A few animal studies confirmed therapeutic potentials of IFN blockade resulting in inhibition of atherosclerotic plaque progression, stabilization of lipid- and macrophage-rich advanced plaques in ApoE knockout mice, and reduction of atherosclerosis in the graft vessels and the aorta in mice heart transplantation models [88,89,90]. Unfortunately, the inhibition of IFN blockade in atherosclerosis has not been tested in humans yet and studies on IFN inhibition in other indications (Crohn diseases and SLE) have been terminated prematurely due to lack of efficacy [87]. Another interesting study compared RA patients and healthy volunteers focusing on kinetics of cholesterol metabolism following a six-week tofacitinib treatment. In the study, the authors observed a reduction of the cholesterol ester fractional catabolic rate with subsequent increasing of HDL cholesterol and LDL cholesterol levels [91].

To summarize, blocking the JAK/STAT pathway may offer some potential to stop/reduce atherosclerosis development. The increment in lipid level did not translate to increased risk for atherosclerosis and is believed to be due to the transposition between several lipid compartments but not due to increased lipid synthesis. Moreover, as we already learned from the study with baricitinib, hypercholesterolemia, which occurred in less than 10% of the patients, was a dose-dependent event. Cholesterol level tends to increase during the first 12 weeks of the treatment, followed by stabilization of total cholesterol, LDL, and HDL serum levels when the treatment was continued [92].

#### 5.2.2. Thromboembolic Events

Data from many clinical trials on JAK inhibitors suggesting increased risk of thromboembolic events in patients treated with JAK inhibitors. This risk is especially high in patients who already have risk factors for venous thromboembolic event (VTE), including history of cardiovascular disease, increased body mass, hypercoagulable states, neoplasm, and history of prior VTE, as well as patients receiving estrogens, patients undergoing major surgery, or patients with movement disabilities. The pathophysiological background of unprovoked thromboembolism, referring to both deep venous thrombosis and pulmonary embolism, is largely unknown, but it is believed to be linked with comorbidities, lifestyle, and risk factors. Recently, however, data on gene expression profiles provided a new insight into the pathogenesis of VTE. Among many mechanisms potentially involved into the progression of the diseases, leukocyte transendothelial migration and JAK-STAT signaling pathway and their related genes were found to be related with recurrent VTE [87]. Moreover, the regulatory network of recurrent VTE displayed that most of differentially expressed genes including intercellular adhesion molecule 1 and protein-tyrosine kinase 2-beta were regulated by STAT3. This may bring many clinical consequences as down regulated JAK/STAT related genes are directly linked to recurrent VTE, so the open question is whether continuation of JAK inhibitor after the first episode of VTE brings the hazard of recurrent VTE [87]. Contrary to this finding, Lu et al. suggested the role of the JAK2-STAT3 pathway in the regulatory network of platelet activation. According to the results from the study, collagen-induced platelet activation was mediated through the activation of JAK2-JNK/PKC-STAT3 signaling. So, the inhibition of this pathway may exert anti-platelet activity [88]. The opposite conclusions come from the study of Ayer et al., who investigated the role of mutation of JAK2 in the development of thrombosis in chronic myeloproliferative diseases, finding no relationship between thromboembolic events and JAK2 mutation [89]. Recently, the role of tissue factor (TF) and its regulatory mechanism attracted attention of many researchers. TFs may be regulated in several ways including this provided by heparanase. Heparanase (heparanase-1) is a mammalian enzyme (endo-β-D-glucuronidase) that degrades side chains of heparan sulfate [90]. Heparanase gene expression is regulated via many pro-inflammatory pathways (cytokines, reactive oxygen species, early growth response 1 transcription factor, and estrogens) and play a role as in the pathogenesis of several inflammatory disorders, such as inflammatory lung injury, cancer development, and chronic colitis [93]. Data from the literature suggests also the role of heparanase in the development of inflammatory arthritis including RA [93]. In line with it, dramatic hyperactivity of heparanase and angiogenesis gene expression in synovium of RA patients were found [94]. Data form patients with thalassemia showed very high activity of heparanase which in turn may activate the coagulation system, leading to thrombotic events. This is partially mediated by a high erythropoietin level which activates the JAK2 pathway. Therefore, the modulation of the JAK2 signaling pathway may help to reduce risk of thrombotic events mediated by heparinase [95]. The other beneficial effect that JAK inhibitors may exert on the coagulation pathway, reducing the risk of thromboembolic events, is the reduction of fibrinogen synthesis in hepatocytes. In this special instance, the effect is indirect and is related to total reduction of inflammatory response via inhibition of IL-6 signaling [96].

The potential mechanisms that lead to an increased risk of thromboembolic event in patients with RA treated with JAK inhibitors remain the subject of controversy. Blocking the JAK/STAT pathway may bring both harmful and beneficial effects depending on the pathophysiological environment. Therefore, it may be suggested that not JAK inhibition alone, but comorbidities and risk factors presented in RA patients working together may in some predisposing patients increase the risk of thromboembolic complications.

## 6. Conclusions

The potent toll to halt inflammatory response has been given to rheumatologists and immunologists. So far, JAK inhibitors showed a high level of efficacy and satisfactorily level of safety. As with all novel compounds many questions arise regarding the safety profile and potential influence of these drugs on the functioning of vitally important internal organs, including the cardiovascular system. With the small synthetic compounds, we may block many cytokines belonging to several families, which exert both pro- and anti-inflammatory potentials. Parallel to this, we may change the expression of hundreds of related genes, creating a new environment for our patients. At the current level of knowledge, the direct influence of JAK inhibitors on the cardiovascular system is neutral or slightly beneficial. As we still accumulate knowledge on this impact, currently available data regarding this influence are not yet conclusive and more studies in this field are required. At this moment, it is still unclear which mechanisms may drive thromboembolic risk and this is the main concern regarding the administration of JAK inhibitors in patients with RA.

## Figures and Tables

**Figure 1 ijms-21-07390-f001:**
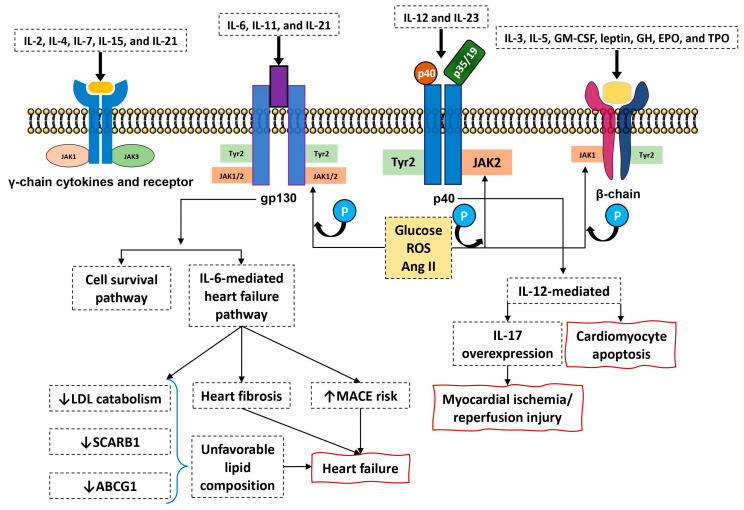
Several cytokine families utilize type I receptors. Receptors with gp130 component transmits signals from IL-6, IL-11, and IL-27 with subsequent activation of JAK1-JAK2 and TYK2 molecules. Cytokines activate (mainly IL-6) heart survival pathway resulting in stabilization of the heart function in ischemia/reperfusion conditions. The same pathway contributes however in deterioration of heart function and increases the risk of major adverse cardiovascular events (MACE), leading to heart fibrosis with subsequent development of heart failure. Cytokines IL-12 and IL-23 that interact with p40 receptor component transmit signals via activation of JAK2 and TYK2 molecules resulting in cardiomyocytes apoptosis. Moreover IL-12 facilitates IL-17 overexpression leading to myocardial/reperfusion injury. Blocking JAK/STAT pathway with JAK inhibitors may therefore result in blocking the heart failure survival pathway but also may reduce MACE incidence and fibrosis of the heart. Blocking of JAK/STAT pathway (mainly IL-6 mediated arm) is also responsible for unfavorable lipids profile changes mediated by reduced LDL catabolism, but this effect may be ameliorated by reduced expression of scavenger receptor class B and ATP-binding cassette G-1. Leading to improvement of lipid composition. Several pathological conditions upregulate JAK/STAT pathway activity (hyperglycemia, reactive oxygen species formation, and angiotensin II) facilitating transmission signals from proinflammatory cytokines.

**Figure 2 ijms-21-07390-f002:**
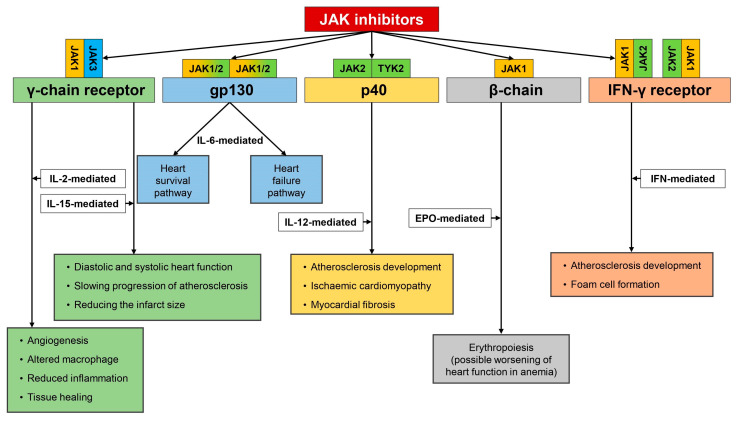
JAK inhibitors targeting JAKs of type I and II receptors. Based on cytokine profile inhibition of JAK/STAT pathway, diverse biological consequences are observed. Inhibition of JAK attached to γ-chain receptor resulting either in beneficial (blocking IL-15, high concentration mediated IL-2 transmission) or detrimental (inhibition of beneficial activity of low IL-2-impaired tissue healing and repair). Inhibition of JAK fused with gp130 receptor reduces IL-6 level. Based on the pathophysiological circumstances, reduced level of IL-6 may contribute to the reduction of heart survival pathway activity or favorably modify heart failure pathway. As far as IL-12 operating via p40 receptor subunit is concerned, inhibition of JAK results in reduction of IL-12-mediated signaling and exerts favorable effects on the cardiovascular system halting progression of atherosclerosis, reducing risk of developing ischemic cardiomyopathy, and myocardial fibrosis. Inhibition of JAK/STAT system transmitting signal from interferon receptor results in reduction of activity of IFN-dependent genes that translates directly to the reduction of foam cell formation and halting progression of atherosclerosis. Finally, some negative consequences may arise as the result of erythropoietin blockade with subsequent anemia development (indirectly contributing to worsening of heart function).

**Figure 3 ijms-21-07390-f003:**
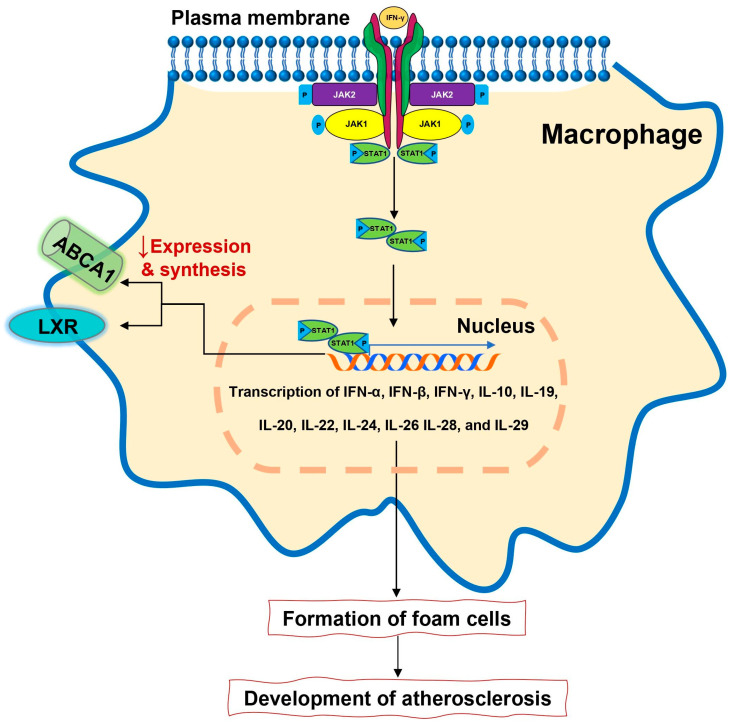
Interferons after ligating to type II receptors activate JAK/STAT pathway resulting in foam cells formation leading directly to atherosclerosis development. That same pathway that transmits signals from interferons contributes to reduced expression of the liver X receptor and decrease synthesis of ATP-binding cassette transporter leading to pro-atherosclerotic lipid composition.

**Table 1 ijms-21-07390-t001:** Currently available classes of disease-modifying antirheumatic drugs (DMARDs).

Typical Drug Representatives	Mode of Action	Side Effects
***csDMARDs***		
Methotrexate	At lower doses (as used in rheumatology) methotrexate inhibits the 5-aminoimidazole-4-carboxamide ribonucleotide transformylase. As a result, it increases extracellular pool of adenosine leading to an overall immunomodulatory activity	Oral ulcers, alopecia, nausea, hepatic and hematologic toxicities, and pneumonitis
***tsDMARDs***		
*JAK inhibitors* ▪ Tofacitinib ▪ Baricitinib ▪ Upadacitinib	Inhibition of JAK molecule and subsequently JAK stat pathway resulting in reducing expression of cytokine related genes	Lipid profile disturbances, higher risk of infections, and thromboembolic complications
***bDMARDs***		
*TNF-α inhibitors* ▪ Infliximab ▪ Etanercept ▪ Golimumab ▪ Adalimumab ▪ Certolizumab pegol	Inhibit (ameliorate) TNF activity upon targeted cells resulting in blockade of inflammatory response driven by this cytokine.	Infections, latent tuberculosis reactivation, neuropathy development (anectodical data), contraindicated in patients with over or latent heart failure, and risk of malignancy
*IL-6 inhibitors* ▪ Tocilizumab	Inhibit IL-6 activity upon targeted cells.	Infections and lipid profile disturbances
*B-cell depletion* ▪ Rituximab	Antibody against B-cell (anti CD-20). Depletion of whole lines of B-cells expressing CD-20 molecule.	Infections, infusion-related reactions, hepatitis B infection reactivation, cytokine released syndrome, and progressive multifocal leukoencephalopathy
*Inhibitors of co-stimulation* ▪ Abatacept ▪ CTLA-4 (CD-152) molecule fused to an immunoglobulin G1 Fc part	CTLA-4 regulates T-cell priming, differentiation, and migration. CTLA-4 ensures homeostasis of regulatory T cells and mediates their immunosuppressive capacity.	serious allergic reactions including anaphylaxis and angioedema, latent tuberculosis reactivation, and higher risk for cancer (i.e., skin cancer)
DMARDs: disease-modifying antirheumatic drugs; csDMARDs: conventional synthetic DMARDs; tsDMARDs: targeted synthetic DMARDs; bDMARDs: biological DMARDs; TNF: tumor necrosis factor; IL: interleukin; CD: cluster of differentiation; CTLA: cytotoxic T-lymphocyte-associated protein.		

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
