# Peer review of "Clinical Aspects of Janus Kinase (JAK) Inhibitors in the Cardiovascular System in Patients with Rheumatoid Arthritis"

_ijms, 2020, doi:10.3390/ijms21197390_

Round 1

Reviewer 1 Report

The review offers an interesting point of view on cardiovascular implications in the use of JAK inhibitors for the treatment of RA. Some aspects could be improved.

Major points

  1. Nowadays, there are three classes of Disease-Modifying Anti-Rheumatic Drugs (DMARD): csDMARD, bDMARD, and tsDMARD. A table reporting the main drug for each class,  the target and adverse effects could be useful
  2. paragraph § 2 “JAK-STAT pathway” : "Compared to the other ones, this paragraph is too long. It clearly explains how the JAK/STAT pathway works, and even if it is very interesting, it is not necessary for the aim of the review. The authors should summarize its content".
  3. In paragraph  § 3 the authors should explain the mechanism through which the JAK/STAT pathway is involved in immunity and autoimmunity
  4. A brief introduction of Jak inhibitors, first-generation and next-generation drugs,  should be included

Minor points

  1. lane 91 “depending” is in capital letter
  2. The quality of figure 1 must be improved
  3. A general figure summarizing how JAK inhibitors are involved in cardiovascular events could  be  suitable

Author Response

Thank you very much for your comments

Please find enclosed answers to your comment

Major comments

1.Nowadays, there are three classes of Disease-Modifying Anti-Rheumatic Drugs (DMARD): csDMARD, bDMARD, and tsDMARD. A table reporting the main drug for each class, the target and adverse effects could be useful

Thank you very much for your comment. We entirely agree that such a table would be useful. We incorporated a new table (Table 1) representing of all the currently available classes of DMARDs.

  1. paragraph § 2 “JAK-STAT pathway” : "Compared to the other ones, this paragraph is too long. It clearly explains how the JAK/STAT pathway works, and even if it is very interesting, it is not necessary for the aim of the review. The authors should summarize its content".

Many thanks for your suggestion. The paragraph has been substantially shortened, some information less irrelevant to the main subject of the paper have been removed (all marked using Track Changes)

  1. In paragraph § 3 the authors should explain the mechanism through which the JAK/STAT pathway is involved in immunity and autoimmunity.

Thank you for your comment. We have provided more data on STAT /JAK system in SLE - a prototype for all autoimmune diseases, focusing on modulation of interferon and possible impact of cells regulating cytokines. Unfortunately, at this moment, activity of JAK/STAT system apart from some direct mutations is assessed in the light of cytokines blocked, therefore, the net effect is dependent upon what cytokines are blocked when a given JAK inhibitor is administered. In detail, the following sentences have been added in the revised manuscript.

SLE is an autoimmune disease with a complex immunopathogenesis where B-cells have been implicated in humoral abnormalities and a prominent type I IFN signature is found in blood of majority of the SLE patients [42]. As the JAK/STAT cascade was identified responsible for the signal transduction from the activated IFN receptor to the nucleus, any disturbance in activity of this pathway may lead to the disease development. Indeed the study on human lupus nephritis (LN) by Arakawa et al. [43] observed increased glomerular staining of STAT3 in renal biopsies of LN patients. Moreover, in patients with different types of glomerulonephritides, STAT3 activation highly correlated with glomerular and tubulointerstitial cell proliferation, interstitial fibrosis, and the level of renal injury.

A brief introduction of Jak inhibitors, first-generation and next-generation drugs, should be included

We entirely agree that JAK inhibitor classification will help the readers to understand our manuscript better. Following your advice, we have added a new paragraph under 4. JAK inhibitors and rheumatoid arthritis. Following sentences have been incorporated.

Based on the selectivity of a given JAK molecule, JAK inhibitors are commonly subcategorized as first and next generation. First generation of JAK inhibitors (i.e., tofacitinib, baricitinib, and peficitinib - approved for RA in Japan) block two or more JAK molecules resulting in inhibition of several cytokines. Consequently, inhibiting a JAK molecule may block more than one pathway, which may in part explain both the drug efficacy and some of the adverse effects observed with JAK inhibitor treatment [47]. In contrast, next generation of JAK inhibitors (i.e., upadacitinib and filgotinib - not approved for RA yet) is characterized by high selectivity thus administration of inhibitors blocks only one specific JAK molecule and selectively inhibits signal from one or limited number of cytokines. This provide the precise mechanism enabling to target one cytokine with one drug.

Minor comments

  1. lane 91 “depending” is in capital letter.

Sentence has been removed.

  1. The quality of figure 1 must be improved.

The quality of all the figures have been improved substantially. Thank you.

Reviewer 2 Report

Here, the authors reviewed the clinical aspects of JNK inhibitors in the cardiovascular system in patients with RA. Their work is well organized. Figures are appropriate and help the readers understand the key message of the review.

Author Response

Thank you very much for your comments